# Evaluation of Antioxidant and Anticancer Activity of Mono- and Polyfloral Moroccan Bee Pollen by Characterizing Phenolic and Volatile Compounds

**DOI:** 10.3390/molecules28020835

**Published:** 2023-01-13

**Authors:** Volkan Aylanc, Samar Larbi, Ricardo Calhelha, Lillian Barros, Feriel Rezouga, María Shantal Rodríguez-Flores, María Carmen Seijo, Asmae El Ghouizi, Badiaa Lyoussi, Soraia I. Falcão, Miguel Vilas-Boas

**Affiliations:** 1Centro de Investigação de Montanha (CIMO), Instituto Politécnico de Bragança, Campus de Santa Apolónia, 5300-253 Bragança, Portugal; 2Laboratório Associado para a Sustentabilidade e Tecnologia em Regiões de Montanha (SusTEC), Instituto Politécnico de Bragança, Campus de Santa Apolónia, 5300-253 Bragança, Portugal; 3Departamento de Química e Bioquímica, Faculdade de Ciências, Universidade do Porto, 4169-007 Porto, Portugal; 4Département de Génies Biologique et Agroalimentaire, Université Libre de Tunis, 30 Avenue Kheireddine Pacha, Tunis 1002, Tunisia; 5Facultad de Ciencias, Universidad de Vigo, Campus as Lagoas, 36310 Vigo, Pontevedra, Spain; 6Laboratory Physiology-Pharmacology and Environmental Health, Faculty of Sciences Dhar El Mehraz, University Sidi Mohamed Ben Abdallah, Fez 30050, Morocco

**Keywords:** antiradical capacity, antitumor activity, bee products, bioactive compounds, phenolic compounds, phenylamides

## Abstract

Bee pollen is frequently characterized as a natural source of bioactive components, such as phenolic compounds, which are responsible for its pharmaceutical potential and nutritional properties. In this study, we evaluated the bioactive compound contents of mono- and polyfloral bee pollen samples using spectroscopic and chromatographic methods and established links with their antioxidant and antitumor activity. The findings demonstrated that the botanical origin of bee pollen has a remarkable impact on its phenolic (3–17 mg GAE/g) and flavonoid (0.5–3.2 mg QE/g) contents. Liquid chromatography–mass spectrometry analysis revealed the presence of 35 phenolic and 13 phenylamide compounds in bee pollen, while gas chromatography–mass spectrometry showed its richness in volatiles, such as hydrocarbons, fatty acids, alcohols, ketones, etc. The concentration of bioactive compounds in each sample resulted in a substantial distinction in their antioxidant activity, DPPH (EC_50_: 0.3–0.7 mg/mL), ABTS (0.8–1.3 mM Trolox/mg), and reducing power (0.03–0.05 mg GAE/g), with the most bioactive pollens being the monofloral samples from *Olea europaea* and *Ononis spinosa*. Complementarily, some samples revealed a moderate effect on cervical carcinoma (GI_50_: 495 μg/mL) and breast adenocarcinoma (GI_50_: 734 μg/mL) cell lines. This may be associated with compounds such as quercetin-*O*-diglucoside and kaempferol-3-*O*-rhamnoside, which are present in pollens from *Olea europaea* and *Coriandrum*, respectively. Overall, the results highlighted the potentiality of bee pollen to serve health-promoting formulations in the future.

## 1. Introduction

The major product of beekeeping activities is known as honey. However, honey bees, the golden insects of nature, can provide a much wider range of products with enormous potential, such as bee pollen, bee bread, propolis, royal jelly, bee venom, and beeswax [1]. This great variety of natural products has been intensively researched and employed in different industries for various purposes [2,3,4]. For example, propolis and bee venom are the subjects of important research in the field of pharmacy due to their strong biological activities [5,6], while honey, bee pollen, and bee bread are considered important products as functional foods due to their nutritional values and remarkable biological activities [7,8,9].

Essentially, bee pollen is the male gametophyte of the plant. Worker bees collect these pollen grains from flowers and mix them with their own secretions, turning them into moist pellets. These pellets then stick to the pollen basket on the hind legs of the bees and begin their journey toward the hive [1]. Reaching the hive, the bees are forced to trespass an apparatus placed at the entrance of the hive, pollen traps, where the pellets are forced to detach from the bees’ legs, following in the trap. The chemical composition and biological activity of bee pollen exhibit significant changes from pollen to pollen [1,7], depending on the type of plant from which this pollen originates, geographical conditions, collection season, as well as storage and processing factors [4,10].

With the development of analytical instruments and methods, the number of studies demonstrating that bee pollen is a natural source of bioactive compounds and micro- and macronutrients has progressed [3,7,10]. These advances have encouraged researchers to evaluate bee pollen, especially for the food sector. For example, there is an intense effort to fabricate functional foods with enhanced nutritional values and more potent biological activities by incorporating bee pollen in different food products, such as bread [11], biscuits [12], and meat [13]. Additionally, several studies stated that bee pollen samples from different geographical locations around the world are a great source of aldehydes, alcohols, fatty acids, phenolic compounds, terpenes, and esters that—when combined—potentiate the pharmaceutical properties of bee pollen, such as antioxidant, anti-inflammation, anticancer, and antidiabetic properties [1,7,14]. Such potential health benefits of bee pollen are particularly linked to the presence of phenolic compounds [1,15]. Studies analyzing the chemical composition of bee pollen verify that it contains several flavonoids (e.g., kaempferol, quercetin, and isorhamnetin,) and flavonoid glycosides among other simple phenolics [4,10,16,17]. Even though this class of compounds is mostly non-nutritive in the diet for humans, there is some evidence to suggest that modest consumption in the long term may reduce the incidence of certain cancers and chronic diseases [18]. Reactive oxygen (ROS) and nitrogen species (RNS) produced owing to the metabolic activity of cells or due to environmental factors can damage biological molecules, such as DNA, enzymes, and cells, and possibly contribute to cellular dysfunction and disease [19,20]. Phenolic compounds have the potential to reduce the adverse effects of ROS and RNS based on various antioxidant action mechanisms. For example, the binding of metal ions needed for catalysis of ROS generation, the scavenging of ROS and RNS or their precursors, the upregulation of endogenous antioxidant enzymes, or the repair of oxidative damage to biomolecules [20]. Additionally, some phenolic compounds react directly with free radicals, quenching them without reacting with other cell components [19,20,21].

It is obvious that more research is needed to reveal food and pharmacological characteristics of bee pollen, leading us to categorize them as mono- and polyfloral and determine the properties of each botanical origin accurately. Even though some countries established standards for bee pollen according to their own national regulations [1], this natural product lacks international standardization. Indeed, it has been emphasized in numerous studies that there is a requirement for more research based on the chemical composition of different bee pollens by referring to their botanical origins [1,2,7,14]. 

To fulfill this lack, we aimed to understand the link between the antioxidant and anti-tumor potentials of mono- and polyfloral bee pollen samples from Morocco depending on the type and abundance of bioactive compounds evaluated by liquid chromatography coupled to diode array detection and electrospray ionization tandem mass spectrometry (LC/DAD/ESI-MS^n^) and gas chromatography–mass spectrometry (GC-MS). In particular, the number of studies that associate the presence of volatile compounds and phenylamides, which represents a significant amount of the bioactive content of bee pollen, with the biological activities is still very limited.

## 2. Results and Discussion

### 2.1. Color and Palynological Assessment

Color and palynological analytical results of bee pollen samples are given in Table 1. Visual inspection of bee pollen loads revealed diversity in color, including light purple, light yellow, orange, yellow, and dark yellow. A notable particularity among the samples was that between BP7 and BP8, the samples had slightly different colors even though they came from the same pollen species. This could most likely be explained by the presence of minor pollen species. However, the oxidation of the samples by exposure to air or light [22] cannot be dismissed.

The botanical origin of the bee pollen samples was easily distinguished under the light microscope according to the morphology of the pollen grains, although, it was challenging to distinguish the species of some pollen grains from the same type because of the high morphological similarity. In these cases, only the genus name was indicated, as in *Coriandrum*, *Carduus*, or *Ononis*. The sample was considered monofloral when the relative frequency of pollen species was ≥80% [23]. The results allowed the classification of five samples as monofloral and three samples as polyfloral. BP1, BP2, BP4, BP5, and BP8 bee pollen samples were assigned to the monofloral classes originated from *Coriandrum* (100%), *Brassica* (90%), *Olea europaea* (100%), *Raphanus* (>80%), and *Ononis* (>95%), respectively. Other samples exhibited pollen species of various botanical origins at different relative frequencies, as in Table 1. These pollens are common to other previously identified in Moroccan bee products, particularly in honeydew honey [24] and bee bread samples (natural fermented bee pollen observed inside the hive) [25]. Evidently, it is quite possible that the harvested bee pollen samples in the same or nearby geographic areas may reveal different botanical origins, as this may vary in line with the dominant flora in the area where the apiaries are located as well as honeybee preference [1].

### 2.2. Total Phenolic and Flavonoid Content 

As shown in Figure 1a, the hydroethanolic bee pollen extracts resulted in a wide range of total phenolic content. The values ranged between 2.7 ± 0.6 and 16.8 ± 1.1 mg of gallic acid equivalents per g of bee pollen (mg GAE/g), with more than six-fold variation. Three extracts exhibited relatively high phenolic contents (>10 mg GAE/g): BP7 (16.8 ± 1.1 mg GAE/g), BP5 (12.1 ± 0.3 mg GAE/g), and BP4 (10.3 ± 0.7 mg GAE/g). In contrast, BP8 and BP1 presented the lowest phenolic contents, with values of 2.7 ± 0.6 and 3.7 ± 0.1 mg GAE/g, respectively. The most notable point among the results was that the samples with the highest and lowest phenolic contents had the same main pollen type, *Ononis*, as given in Table 1. However, while sample BP8 is monofloral with *Ononis* (>95%), sample BP7 is polyfloral and contains other pollen types, such as *Lythrum* or *Acacia*, which may be responsible for the increment in the phenolic content.

The flavonoid content of bee pollen samples was measured by the aluminum chloride method, which is commonly employed to determine the amount of flavonoids, and the results were illustrated in Figure 1b. The highest flavonoid content was recorded in the BP6 extract (3.2 ± 0.1 mg of quercetin equivalents per g of bee pollen (mg QE/g), which was dominated by *Helianthemum* (>78%) from the Cistaceae family. The BP6 was followed by BP4, BP7, and BP5 extracts, with values of 2.1 ± 0.1, 1.7 ± 0.2, and 1.4 ± 0.2 mg QE/g, respectively. Among the analyzed samples, these three samples demonstrated the highest values of total phenolic compounds with a positive correlation. Concerning samples BP1, BP2, and BP3, a decrease in the flavonoid content was observed in contrast with the increase in the total phenolic content, Figure 1a,b. The findings revealed that high phenolic content may not always correlate with high flavonoid content, as stated by some authors before [26].

The total phenolic or flavonoid values of the tested bee pollen were significantly (*p* < 0.05) different from each other in multiple comparisons, with one exception (BP1-BP8 in flavonoid content). Our trends are similar to the findings of Morais et al. [27] and Araújo et al. [28], who stated that the total phenolic (from 10.5 to 16.8 mg GAE/g; *n* = 5) and flavonoid content of bee pollen samples (from 1.4 to 9.1 mg QE/g; *n* = 9) could lead to variable values depending on the pollen species. Additionally, the present results are also consistent with other studies previously reported for bee pollen at different geographic locations [12,17,29].

### 2.3. LC/DAD/ESI-MS^n^ Bioactive Compounds Analysis

The optimized chromatographic conditions provided the identification and quantification of the bioactive compounds in the Moroccan bee pollen samples. The ESI source in negative ion mode was chosen for the assessment of the compounds, and the most intense peak in MS was selected as the precursor ion (*m/z*). The compound identification was performed according to the detected precursor ion and MS/MS fragmentation by comparison with standards and reported data in the literature. When this information was not available, the MS data were validated by combining the described UV (ultraviolet) spectra and retention time data available in the literature. The quantification was performed through the chromatogram at 280 nm and using the calibration curves of the phenolic compound with the closest structural similarity.

The chromatographic profile allowed the identification of a total of 48 bioactive compounds in the bee pollen samples, of which 31 were flavonoids, mostly flavonol glycosides; 13 were phenylamide compounds; and 4 were phenolic acids, as in Table 2 and Appendix A.

Myricetin, quercetin, isorhamnetin, kaempferol, and herbacetin glycosides were the most abundant flavonoids identified, consistent with previous studies reporting them as the main phytochemical compounds in bee pollen of various botanical origins [1,4,14].

Bee pollen flavonol aglycones presented a series of losses associated to different sugar moieties, such as pentosides, hexosides, deoxyhexosides, and deoxyhexosyl-hexosides. The peaks corresponding to acetyl glycosides and malonyl glycosides were observed in several compounds. Peak 6 presented a pseudomolecular ion [M-H]^−^ at *m/z* 667, releasing an MS^2^ fragment at *m/z* 316 ([M-H-350]^−^, loss of an acetyl deoxyhexosyl-hexoside moiety), corresponding to myricetin, as in Figure 2 and Figure 3a. Additionally, Peak 14 was identified as a myricetin derivate with a [M-H]^−^ at *m/z* 565 and presented a fragmentation pattern with an MS^2^ with an ion at *m/z* 521 formed by the loss of a carboxyl group (−44u). The following MS^3^ spectrum indicated a loss of an acetyl hexoside moiety (−204u). The compound was tentatively identified as myricetin-*O*-malonyl hexoside, as in Figure 2 and Figure 3b. A similar fragmentation pattern was observed for other flavonol glycoside derivatives, such as quercetin-*O*-malonyl deoxyhexosyl-hexoside (peak 20, *m/z* 695), isorhamnetin-*O*-malonyl pentosyl-hexoside (peak 25, *m/z* 695), 3′,4′,5′,3,5,6,7-heptahydroxy-flavonol-*O*-malonyl hexoside (peak, 27, *m/z* 579), and kaempferol-*O*-malonyl rutinoside (peak 31, *m/z* 533), as in Figure 2 and Table 2 and Appendix A.

Compounds such as as quercetin-*O*-diglucoside, kaempferol-*O*-diglucoside, and kaempferol-3-*O*-rutinoside—assigned to the precursor ions [M-H]^−^ at *m/z* 625, *m/z* 609, and *m/z* 593—were the most common flavonoid glycosides within all samples. These compounds have been reported in Portuguese bee pollen, which had predominantly *Plantago* sp., *Crepis capillaris*, and *Cytisus striatus* pollen species [10].

Comparing the samples individually, BP6 exhibited a profile with more diversity of phenolic compounds and the highest total concentration (8.8 mg/g), which was in accordance with the previous results in the flavonoid content of it. A similar situation was observed for the BP3 sample concerning the diversity of compounds. This could be related to the polyfloral nature of BP3 and BP6, for which different plant sources contribute to the high diversity of compounds. On the other hand, BP2, with 60% *Brassica* pollen, was the poorest in terms of compounds. Chromatographic results also revealed that Moroccan bee pollen samples contained four phenolic acids—caffeic acid (*m/z* 179), caffeic acid hexoside (*m/z* 341), *p*-coumaric acid (*m/z* 163), and *p*-coumaric acid hexoside (*m/z* 325)—at low concentrations in BP1, BP2, BP5, and BP8. Such phenolic acids have a wide distribution in the plant kingdom, and their presence has been associated with biological activities such as antiproliferative, antioxidant, and antimicrobial activities [30,31].

The employed chromatographic method allowed the identification of another group of chemical compounds, phenylamides. Even though these compounds have not been the subject of research often, they are responsible for some functions in plants. Phenylamides exist in high concentrations in higher plants, especially on the surface of male reproductive organs, namely pollen [32]. The reason for this is still a mystery, yet some researchers have stated that these compounds may be related to the protection of plant genetic material inside pollen grains from UV light [32]. Regardless, these compounds are obviously a major component of pollen grains, including bee pollen [4,17]. Phenylamides are molecular products chemically formed via covalent bonds between the carboxylic groups of hydroxycinnamic acids (e.g., coumaric acid, ferulic acid, and caffeic acid) and amine groups of aliphatic di- and polyamines or aromatic monoamines [33].

Phenylamide compounds were not detected in samples BP1, BP2, and BP4, as shown in Table 2, which can be due to the low concentration of these compounds in the samples or be related to the applied extraction method. The rigid pollen double-layer can have meaningful effects on the recovery of compounds as previously discussed when applying different extraction techniques [17]. Among the peaks detected in bee pollen phenolic extracts, all phenylamides showed specific UV spectra with a UVmax at around 298 and 310 nm, [4], as in Table 2 and Appendix A.

Confirmed with: ^a^ MS^n^ fragmentation; ^b^ Standard; References: ^c^ Kang et al. [34]; ^d^ El Ghouizi et al. [4]; ^e^ Aylanc et al. [10]; ^f^ Sobral et al. [35]; ^g^ Llorach et al. [36]; ^h^ Falcão et al. [37]; ^i^ Mihajlovic et al. [38]. BP: bee pollen. nd: not detected.

Moroccan bee pollen contained several phenylamides, such as *N*¹-*p*-coumaroyl-*N*⁵; *N*¹⁰-dicaffeoylspermidine (*m/z* 614); *N*¹, *N*⁵, *N*¹⁰-tri-*p*-coumaroylspermidine (*m/z* 582) and its four isomers; *N*¹, *N*⁵-di-*p*-coumaroyl-*N*¹⁰-caffeoylspermidine (*m/z* 598); and tetracoumaroyl spermine (*m/z* 785) and its five isomers. *N¹*, *N⁵*, *N¹⁰*-*tri*-*p*-coumaroylspermidine was the most common compound among the samples, with a concentration of 10.5 ± 0.1 mg/g in BP7, implying approximately a 10-fold difference compared to the average of other samples. Previously [17], this phenylamide was described in high concentrations in bee pollen samples containing mainly *Jasione montana* (Campanulaceae family), *Eucalyptus* (Myrtaceae), and *Rubus* (Rosaceae). Another common compound in the bee pollen samples was the tetracoumaroyl spermine and its isomers, present in BP3 and BP6 samples with concentrations ranging from 0.14–0.23 mg/g. 

### 2.4. Volatile Compounds Profiling

Mono and polyfloral bee pollen volatile compounds were extracted using the solid phase microextraction (SPME) technique followed by GC-MS analysis. The quantification was obtained directly from the total ion chromatogram (TIC) and expressed as a relative percentage. Linear retention indices (LRI) were calculated for each component detected. The list of volatile compounds with the calculated LRI and relative concentration (R%) is given in Table 3 and Appendix A. Moroccan bee pollen presented a wide variety of volatile compounds, with a total of 47 compounds identified, which included 13 aldehydes, 12 esters, 5 hydrocarbons, 5 ketones, 5 terpenes (3 oxygen-containing monoterpenes, 1 monoterpene hydrocarbon, and 1 sesquiterpene hydrocarbon), 4 carboxylic acids, and 1 ether. The great diversity in the composition is due to the different botanical origins, but may also be influenced by the harvesting time, conservation methods, and extraction methodology [39].

Generally, the most common and abundant compounds were hexanal, ranging from 5.3 to 59.8%, and 3,5-octadien-2-one and its isomer, with a concentration ranging from 2.6 to 27.5%. Additionally, octanoic acid was present in all samples in a range from 1.9 to 7.7%, with the exception of the BP8 sample. 3,5-octadien-2-one (22.9%) and octanal (16.6 %) were the most dominant compounds in *Coriandrum* monofloral bee pollen (BP1). In total, six volatile organic compounds were identified in BP2, and 2,4-heptadienal (33.6%) and hexanal (14.9%) from the aldehyde group were detected in high percentages. Additionally, 3,5-octadien-2-one (27.5%) from the ketone group and eucalyptol (10.6%) from the oxygen-containing monoterpenes were found in high concentrations. Volatile compounds, such as hexanal and octanal, were previously identified as being common in different bee pollen samples from Croatia [40], and the presence of eucalyptol was also detected at a relatively low rate (1.9%) in bee pollen samples from Latvia [29]. BP3 sample showed a rich profile in fatty acids and their esters, including hexanoic acid (20.3%), ethyl decanoate (16.7%), ethyl octanoate (14.8%), and methyl octanoate (11.5%). Various organic compounds from different classes were present in the BP4 and BP5 samples, in which methyl octanoate (13.1%) and hexanoic acid (20.5%) were the main compounds in each sample, respectively. Volatile organic compound results reported for polyfloral bee pollen samples (*n* = 16) in a study conducted by Prudun et al. [40] revealed the presence of these two compounds, and yet the samples had different botanical origins than those described in the current study. As previously described for the other samples, BP6 and BP7 presented high concentrations of 3,5-octadiene-2-one, with values of 12.4% and 23.9%, respectively. Hexanal was also identified as one of the main compounds in BP6 (12.3%), BP7 (21.0%), and BP8, which showed a relative percentage of 59.8%. This revealed that *Ononis* bee pollen is a good source of aldehydes. Some authors previously stated that hexanal is biologically active, emphasizing its significant effect on the inhibition of microbiological contaminants [41].

Although volatile compounds are not frequently associated with properties such as antioxidant and antitumor, which we discuss in the next section, they are known to have some biological activities, and revealing their presence may significantly affect consumers’ preferences due to factors such as taste and aroma when bee pollen is used as a food supplement and food ingredient [29,40]. It is therefore important to identify the compounds present by referring to the botanical origin of bee pollen.

### 2.5. Biological Activity

#### 2.5.1. Antioxidant Capacity

The antioxidant capacities of mono- and polyfloral Moroccan bee pollen samples were measured by DPPH (2,2-diphenyl-1-picrylhydrazyl), ABTS [2,2′-azinobis-(3-ethyl-benzothiazoline-6-sulfonic acid)], and reducing power assays and the results are shown in Figure 4a–c.

DPPH radical scavenging activities ranged from EC_50_ 0.71 mg/mL to EC_50_ 0.28 mg/mL, which indicates a 2.5-fold change. Here, a low EC_50_ value indicates high radical scavenging activity. The monofloral sample BP7 exhibited the highest antioxidant activity (0.28 mg/mL) together with *Olea europaea* monofloral bee pollen (BP4), followed by BP6 (0.29 mg/mL), BP3 (0.41 mg/mL), BP5 (0.45 mg/mL), and BP2 (0.52 mg/mL).

In the ABTS assay, the radical scavenging values ranged between 0.81 and 1.26 mM Trolox equivalents per mg of bee pollen, which represent a lower (approximately 1.6-fold) variation compared to the DPPH. As in the DPPH assay, here the BP7 sample showed the highest antioxidant activity with a value of 1.26 mM Trolox/mg, followed by BP6 (1.10 mM Trolox/mg), BP5 (1.01 mM Trolox/mg) and BP4 (0.97 mM Trolox/mg). Along with this, BP1 and BP8, representing monofloral bee pollen samples of *Coriandrum* and *Ononis*, respectively, had the lowest activity in both radical scavenging tests.

The results of the reducing power assay to measure the reduction potential (Fe^3+^→Fe^2+^) of bioactive compounds in bee pollen samples were slightly different from the other two antioxidant assay findings. Among the samples, high reducing power activity was measured as 0.05 mg GAE/g in BP2 and BP8, while the lowest value was measured as 0.03 mg GAE/g in BP1, BP3 and BP4. The remaining samples showed an antioxidant capacity of 0.04 mg GAE/g.

Researchers agree that a single method is often not sufficient to quantify the potential activities of antioxidants, so employing antioxidant quantification assays based on different working principles is a necessary method of comprehensively evaluating the material under analysis [17,31,42]. Antioxidant results obtained from the current study demonstrated that some samples, such as BP4, BP5, BP6, and BP7, contain pollen species with potent free radical scavengers with minor reducing power activity. The antioxidant potential of the samples could be attributed to their total bioactive compound content, especially to the phenolics [10,29]. The phenolic and flavonoid contents and the calculated total amount of phenolic compounds from the LC-DAD-ESI-MS^n^ analysis were highest in the BP4, BP5, BP6, and BP7 bee pollen samples, which were those that exhibited higher radical scavenging profiles in the DPPH and ABTS assays. This situation indicated the existence of a correlation between the amount of phenolic compounds and antioxidant capacity, with a strong relationship between radical scavenging activity and total phenolic content (Figure 4d,e), as well as a moderate correlation with flavonoid content (Figure 4g,h). Dudonné et al. [42] previously highlighted that the phenolic content determined using Folin–Ciocalteu analysis correlated with DPPH and ABTS, showing stronger free radical inhibition values in parallel with the increase in phenolic content. Our results are also consistent with those previously reported for the antioxidant activities of bee pollen from various geographical locations, such as Brazil, Poland, Lithuania, and China [12,28,29]. The high correlation found may be due to the presence of several flavonols, such as quercetin, kaempferol, and myricetin derivatives, with a planar structure caused by the hydroxyl group in position 3 that promotes a higher radical capture due to easier conjugation and electron delocalization. The high number of hydroxyl groups is another factor contributing for the potency of those compounds as electron scavengers. It should be noted that the negative correlation of phenolic and flavonoid content with DPPH was due to the expression of DPPH as EC_50_, where a low value corresponds to a high radical scavenging activity.

On the other hand, both mono- and polyfloral bee pollen did not exhibit any appreciable antioxidant activity in the reducing power test, although there were statistically significant (*p* < 0.05) differences among their phenolic compound concentration, as in Figure 1. In the correlation analysis, there is no significant (*p* > 0.05) link between the phenolic/flavonoid content and reducing power, as shown in Figure 4f,i. This could be due to the stronger activity of individual compounds present in the sample rather than being related to the high phenolic content.

#### 2.5.2. Antitumor Activity

Each bee pollen extract was screened for potential in vitro cytotoxicity activity against human-cancer-derived cell lines, such as stomach gastric adenocarcinoma (AGS), epithelial colorectal adenocarcinoma (CaCo2), cervical carcinoma (HeLa), breast adenocarcinoma (MCF-7), and non-small-cell lung cancer (NCI-H460) as well as a non-tumor cell line, human fetal osteoblastic (hFOB). The growth inhibition (GI) of the cells was not significant in most of the samples (GI_50_ > 1000, μg/mL). BP1 (734 ± 7 μg/mL) and BP4 (495 ± 6 μg/mL) showed cytotoxicity effects exclusively against MCF-7 and HeLa, respectively, Figure 5.

The cytotoxic activities of plant-based materials are generally associated with the presence of phenolic compounds in their chemical composition. Along with this, it is known that the types and concentrations of these natural compounds have a determinative role in the inhibition of growth, proliferation, and invasion of cancer cells in different pathways. Ravishankar et al. [43] mentioned the ability of quercetin in the downregulation of oncogene expression as well as the upregulation of tumor suppressor genes. Regarding BP1, the detected compound at a distinctive concentration was kaempferol-3-*O*-rhamnoside (1.60 mg/g), and this flavonol was previously reported by Lee et al. [44] to suppress the protein expression and metastasis-promoting markers of MCF-7 breast cancer cells, thereby reducing their migration and invasion ability to the level of control. BP4 showed cytotoxic activity against the HeLa cell line with a higher inhibition rate than BP1. Even though the monofloral BP4 from *Olea europaea* species did not actually show a remarkable profile in terms of bioactive compound diversity, the chromatographic result revealed its richness in quercetin-*O*-diglucoside (3.3 ± 0.1 mg/g) compound, which might have determined its main action against the HeLa cancer cell line, as referred to in a previous study above. Additionally, none of the tested bee pollen fractions exhibited any cytotoxic activity against hFOB employed as the normal cell line.

## 3. Materials and Methods

### 3.1. Collection and Preparation of Bee Pollen

Bee pollen samples were collected by local beekeepers between 2015 and 2017 from different locations in Morocco, Table 1, and stored in the freezer (−20 °C) until further analysis.

### 3.2. Palynological Analysis

Palynological analysis was performed according to a method previously described [45]. Accordingly, 10 mL of distilled water was added to 1 g of bee pollen samples and vortexed vigorously. Then, a 100 µL aliquot was placed on a slide, and after drying, one drop of glycerin jelly was added for permanent preparation. Pollen grain identification was performed by optical microscope. A reference collection from the botanical laboratory of the University of Vigo, Spain, and different pollen morphology guides were used for the identification of pollen types. The relative frequency of each pollen type was calculated by counting a minimum of 500 pollen grains per slide.

### 3.3. Extraction of Phenolic Compounds

Aliquots of 2 g bee pollen samples were accurately ground and weighed into a centrifuge tube and extracted with 15 mL of ethanol/water (70:30, *v*/*v*) at 70 °C for 30 min, in a water bath at 100× *g*. The mixture was vacuum filtered, and the derived extract was stored at −20 °C until further analysis.

### 3.4. Total Phenolic and Flavonoid Content

The total phenolic compounds in the bee pollen samples were quantified spectrophotometrically according to a previously reported method [10]. Briefly, 0.5 mL of the extract (5 mg/mL) was mixed with Folin–Ciocalteu’s reagent (0.25 mL). After 3 min, 1 mL of 20% sodium carbonate was added to the mixture, and the volume was made up to 5 mL with distilled water. The solution was kept at 70 °C for 10 min and then cooled in the dark at room temperature for 20 min. Subsequently, the mixture was centrifuged for 10 min at 5000× *g,* and the absorbance was measured at 760 nm (Analytikijena 200–2004 spectrophotometer, Analytik Jena, Jena, Germany). The total phenolic content was expressed as mg GAE/g bee pollen (GAE—gallic acid equivalents).

The aluminum chloride method was carried out to determine the total flavonoid content [10]. An aliquot of 0.2 mL of sample solution (5 mg/mL) was mixed with 0.2 mL of aluminum chloride solution (2% AlCl_3_ in acetic acid/methanol, 5/95, *v*/*v*). Following this, 2.8 mL of methanol with 5% glacial acetic acid was added. The mixture was then incubated in the dark at room temperature for 30 min, and the absorbance was measured at 415 nm using a spectrophotometer. The total flavonoid content was expressed as mg QE/g bee pollen (QE—quercetin equivalents).

### 3.5. LC/DAD/ESI-MS^n^ Analysis

The samples for analysis were prepared according to the method previously described [17]. Briefly, 20 mg of bee pollen extract was dissolved in 2 mL of 80% ethanol, filtered through a 0.22 μm membrane, and kept in the freezer at −32 °C until analysis.

A Dionex UltiMate 3000 ultra-pressure liquid chromatography instrument connected to a diode array and attached to a mass detector was used for LC/DAD/ESI-MS^n^ analyses (Thermo Fisher Scientific, San Jose, CA, USA). LC was run in a Macherey–Nagel Nucleosil C18 column (250 mm × 4 mm id; particles diameter of 5 mm, end-capped), and its temperature was kept constant at 30 °C. The conditions applied in the liquid chromatography were based on previous work [4]; the flow rate was 1 mL/min, and the injection volume was 10 μL. The final spectra data were accumulated in the wavelength interval of 190–600 nm. The results were expressed as mg/g of pollen. The mass spectrometer was operated in the negative ion mode using Linear Ion Trap LTQ XL mass spectrometer (Thermo Scientific, CA, USA) equipped with an ESI source. The source’s voltage was 5 kV, in addition to −20 V and −65 V for the capillary and the tube lens, respectively. The capillary’s temperature was fixed to 325 °C. Both sheath and auxiliary gas (N_2_) flows were fixed to 50 and 10 (arbitrary units). Mass spectra were acquired by full range acquisition covering 100–1000 *m/z*. For the fragmentation study, a data-dependent scan was performed by deploying collision-induced dissociation (CID). The normalized collision energy of the CID cell was set at 35 (arbitrary units). Data acquisition was carried out with the Xcalibur^®^ data system (Thermo Scientific, San Jose, CA, USA).

Quantification was achieved using calibration curves for *p*-coumaric acid (0.00925–0.4 mg/mL; y = 2.06 × 10^7^x − 3.5 × 10^5^; *R*^2^ = 0.973), kaempferol (0.037–1.6 mg/mL; y = 4.27 × 10^6^x + 1.98 × 10^5^; *R*^2^ = 0.983), chrysin (0.0185–0.8 mg/mL; y = 1.20 × 10^7^x − 5.83 × 10^4^; *R*^2^ = 0.999), quercetin (0.037–1.6 mg/mL; y = 3.9 × 10^6^x + 4.65 × 10^5^; *R*^2^ = 0.937), naringenin (0.0185–0.8 mg/mL; y = 7.85 × 10^6^x − 3.04 × 10^5^; *R*^2^ = 0.978). When the standard was not available, the compound quantification was expressed in the equivalent of the structurally closest compound. The results were expressed as mg/g of pollen.

### 3.6. Volatile Compounds Analysis

#### 3.6.1. Solid Phase Microextraction

Preliminary optimization of the extraction time and the addition of saline solution led to specific extraction conditions. Approximately 2.5 g of ground bee pollen was mixed with 2.5 mL of a 30% sodium chloride solution in a glass bottle until homogenization. The vial was sealed with a predrilled septum and placed in a thermostatic bath at 50 °C. Headspace sampling was performed using a manual SPME holder equipped with a 65 µm polydimethylsiloxane/divinylbenzene (PDMS/DVB) StableFlex fiber (Supelco, Bellefonte, PA, USA). Sampling of the volatile bee pollen compounds was achieved by inserting the fiber through the septum and exposing it to the headspace for 60 min with continuous stirring. The fiber was then retracted and transferred to the injector port of the gas chromatograph where the compounds were desorbed for 5 min.

#### 3.6.2. GC-MS Profiling and Quantification

The volatile compounds analysis was carried out according to a previously reported method with some modifications [46]. The GC-MS unit consisted of a Perkin Elmer system (GC Clarus^®^ 580 GC module and Clarus^®^ SQ 8 S MS module) gas chromatograph equipped with DB-5 MS fused-silica column (30 m × 0.25 mm i.d., film thickness 0.25 μm; J & W Scientific, Inc.) and interfaced with a Perkin-Elmer Turbomass mass spectrometer (software version 6.1, Perkin Elmer, Shelton, CT, USA). The SPME fiber was desorbed at 250 °C for 5 min. The oven temperature was programmed as 40–170 °C, at 3 °C/min, subsequently at 25 %/min up to 290 °C, and then held isothermal for 15 min. The transfer line temperature was 280 °C; ion source temperature, 230 °C; carrier gas, helium, adjusted to a linear velocity of 40 cm/s; ionization energy, 70 eV; scan range, 40–300 u; scan time, 1s. Identifications were based on the comparison of the obtained spectra with those of the NIST mass spectral library and were confirmed using linear retention indices determined from the retention times of an *n*-alkane (C_7_–C_36_) (Supelco, Bellefonte, PA, USA) mixture analyzed under identical conditions. They were compared to published data and, when possible, to commercial standard compounds. Quantitation (average value for three replicates per sample) was carried out using relative values directly obtained from peak TIC.

### 3.7. Antioxidant Capacity of Bee Pollen

Three different assays based on different working mechanisms were employed to measure the antioxidant capacity of bee pollen samples.

#### 3.7.1. DPPH Free Radical Scavenging Activity

DPPH free radical scavenging activity of bee pollen was performed according to Aylanc et al. [10]. A volume of 0.15 mL of the phenolic extract solutions, with concentrations ranging from 0.034 to 0.5 mg/mL were mixed with 0.15 mL of DPPH (0.024 mg/mL) and the absorbance was read at 515 nm using an ELX800 Microplate Reader (Bio-Tek Instruments, Inc., Winooski, VT, USA). The percentage of radical inhibition was calculated using the following equation:(1)$$\%\;Inhibition=[(A_{DPPH}-A_{Sample})/A_{DPPH}]\;\times \;100$$

The amount of antioxidant required to decrease the initial DPPH concentration by 50% (EC_50_) was achieved by plotting the inhibition percentage against the extract concentration.

#### 3.7.2. ABTS Free Radical Scavenging Activity

The ABTS assay was carried out to determine the ability of bee pollen samples to scavenge the ABTS radical cation using Trolox as the standard, according to the previously described method with modifications [47]. Stock ABTS^+^ solution was prepared from 7 mM ABTS and 2.45 mM sodium persulfate in deionized water. The ABTS^+^ solution was diluted with distilled water to obtain an absorbance of 0.700 (±0.020) at 734 nm. Bee pollen extract (5mg/mL, 0.04 mL) was added to the diluted ABTS^+^ solution (0.96 mL) and mixed immediately. The mixture was incubated for 10 min in the dark, and the absorbance was determined at 734 nm. The percentage of inhibition was calculated by the formula:(2)$$\%\;Inhibition=\frac{\left(1-A_{Sample}\right)}{A_{Control}}\times 100$$

*A**_Sample_* is the absorbance of ABTS^+^ solution with sample, and *A**_Control_* is the absorbance of ABTS^+^ solution without sample. The Trolox equivalent antioxidant capacity of the bee pollen samples (mM Trolox/mg bee pollen extract) was calculated using the calibration curve as follows:(3)$$\mathrm{TEAC}\;\left(mM\frac{Trolox\;}{mg}\;bee\;pollen\;extract\right)=\frac{\left(\%\;InhibitionSample-b\right)\;}{a}\times \frac{aliquot\;volume\;\left(mL\right)}{\;bee\;pollen\;weight\;\left(mg\right)}$$
where *a* and *b* are the slope and the intercept of the calibration curve, respectively.

#### 3.7.3. Reducing Power

The reducing power assay was performed according to a previously reported method [10]. Bee pollen extract (5 mg/mL, 0.25 mL) was mixed with sodium phosphate buffer (pH = 6.6, 1.25 mL). Potassium ferricyanide (1%, 1.25 mL) was added, and the mixture was incubated at 50 °C for 20 min. Then, trichloroacetic acid (10%, 1.25 mL) was added. The mixture was centrifuged at 3000× *g* for 10 min, and 1.25 mL was removed from the top to a new tube. Following, 1.25 mL of water and 0.25 mL of 0.1% ferric chloride were added, and the absorbance was read at 700 nm. The values were expressed as mg GAE/g bee pollen.

### 3.8. Cytotoxic Activity

To evaluate the cytotoxic activity of bee pollen extract with the Sulforhodamine B (SRB) colorimetric assay [48], 5 human tumor cell lines were used: AGS, CaCo2, HeLa, MCF-7, and NCI-H460 as well as hFOB a non-tumor cell line. The treatment solution was prepared from a 20 mg/mL hydroethanolic bee pollen extract, which was freeze-dried and then diluted to various concentrations (125 μg/mL to 2000 μg/mL).

The cell lines subcultures were performed in RPMI-1640 medium enriched with 2 mM glutamine, 100 U/mL penicillin, 100 μg/mL streptomycin, 10% FBS and kept in a humidified air incubator containing 5% CO_2_ at 37 °C. After 24 h of incubation, the attached cells were treated with different extract concentrations and incubated again for 48 h. Afterward, the adherent cells were fixed with cold trichloroacetic acid (TCA 10%, 100 μL) and incubated at 4 °C for 1 h. Subsequently, the cells were washed with deionized water and dried. SRB solution (SRB 0.1% in 1% acetic acid, 100 μL) was added to each plate well and incubated for 30 min at room temperature. The unbound SRB was removed with 1% acetic acid, and the plates were air-dried. The bound SRB was solubilized with Tris (10 mM, 200 μL). To measure the absorbance at 540 nm, an ELX800 microplate reader (Bio-Tek Instruments, Inc; Winooski, VT, USA) was used. Elipticine was used as a positive control, and the results were expressed as GI_50_ values in μg/mL (sample concentration that inhibited 50% of the net cell growth).

### 3.9. Statistical Analysis

All analyses were performed in triplicate, and the results were denoted as mean ± standard deviation (SD). The obtained data were analyzed using GraphPad Prism version 9.4 (San Diego, CA, USA). One-way analysis of variance, followed by Tukey’s multiple comparison test, was conducted to see whether there is statistical significance. *p* < 0.05 was considered significant. Additionally, Pearson’s correlation coefficients were calculated to ascertain the relationship between the tested parameters.

## 4. Conclusions

Herein, Moroccan mono- and polyfloral bee pollen samples were subjected to different tests to determine their antioxidant and antitumor potential after evaluating the individual volatile compounds and their amounts as well as the phenolic and flavonoid contents of the samples. Several differences were found between monofloral and polyfloral bee pollen samples in terms of both diversity and concentration of bioactive compounds. Regardless of mono- and polyfloral classes, all bee pollen samples showed significant activities in free radical scavenging tests, but they did not show significant performance in reducing power and antitumor tests, except with some minor activities. The monofloral samples BP4 from *Olea europaea* and BP7 from *Ononis spinose* showed the highest radical scavenging activity against DPPH and ABTS, respectively, while BP2 and BP8 equally showed the highest reducing power activity. Moreover, among the samples tested, only BP1 (against MCF-7) and BP4 (against HeLa) showed cytotoxicity activity, which may be linked to the presence of specific flavonoids such as quercetin-*O*-diglucoside and kaempferol-3-*O*-rhamnoside. The antioxidant action of bee pollen samples and their cytotoxic effects on some cancer cells may be summed concisely as a combination of their phenolic, phenylamide, and volatile compounds content. Overall, the findings of our study contribute to establishing quality standards for Moroccan bee pollen and promoting the consumption of this natural beehive product, with potential evidence for the prevention or reduction of some health problems in which free radicals play major roles.

## Figures and Tables

**Figure 1 molecules-28-00835-f001:**
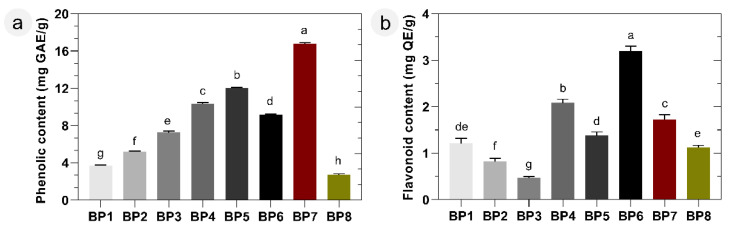
(**a**) Phenolic content and (**b**) flavonoid content of mono- and polyfloral bee pollen samples. Different letters (a–h) indicate significant differences on the phenolic content (*p* < 0.05).

**Figure 2 molecules-28-00835-f002:**
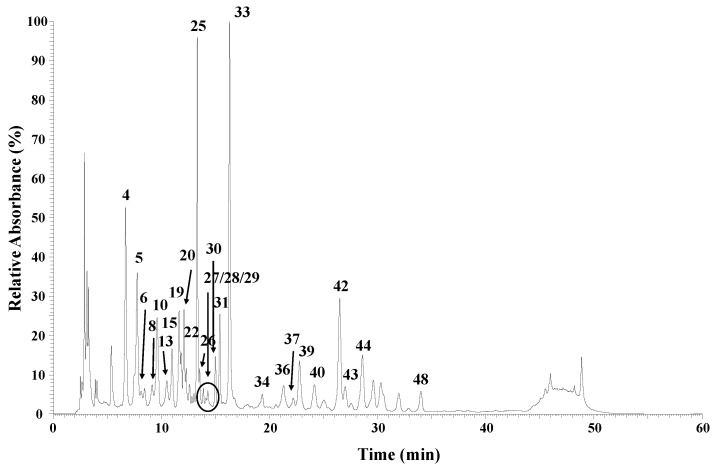
Chromatographic profile at 280 nm for BP6 phenolic extract.

**Figure 3 molecules-28-00835-f003:**
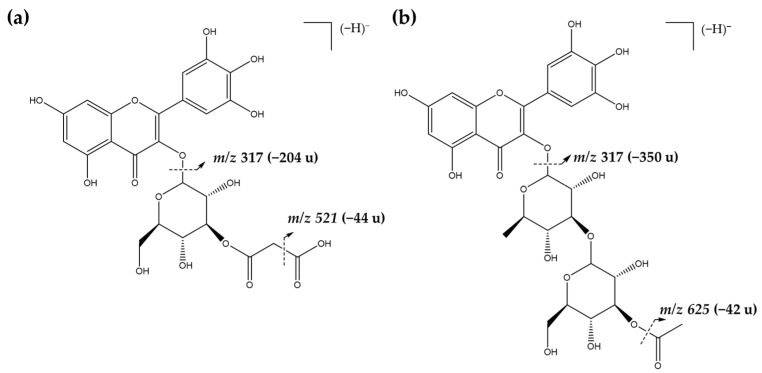
Mass fragmentation pattern for the tentative identification of (**a**) myricetin-*O*-acetyl deoxyhexosyl-hexoside and (**b**) myricetin-*O*-malonyl hexoside present in sample BP6.

**Figure 4 molecules-28-00835-f004:**
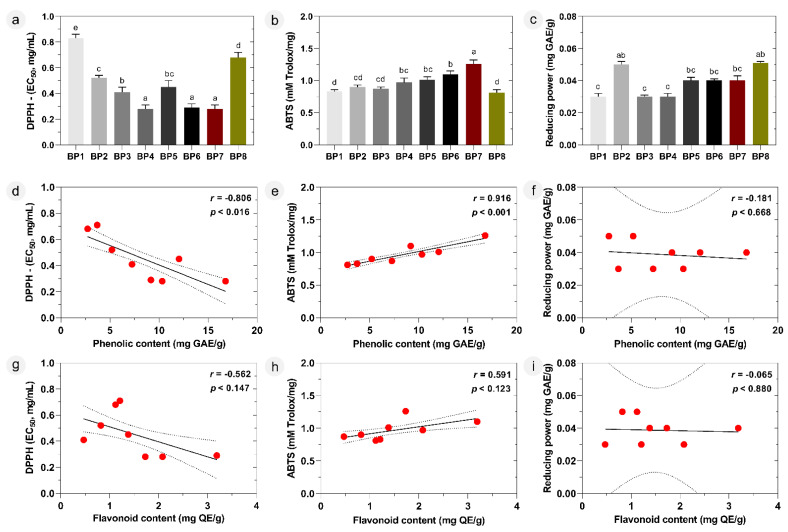
Antioxidant activity of mono- and polyfloral bee pollen samples. (**a**) DPPH free radical scavenging activity; (**b**) ABTS free radical scavenging activity; (**c**) reducing power activity and correlation of (**d**,**g**) DPPH; (**e**,**h**) ABTS; and (**f**,**i**) reducing power activity with phenolic content and flavonoid, respectively. Different letters (a–e) mean significant differences (*p* < 0.05). *r*: Pearson’s correlation coefficient.

**Figure 5 molecules-28-00835-f005:**
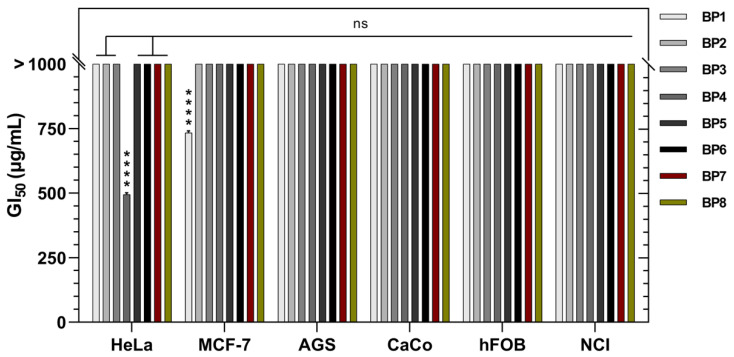
Cytotoxicity activity of the Moroccan bee pollen samples. ****: *p* < 0.0001.

**Table 1 molecules-28-00835-t001:** Geographical location, codes, colors, and botanical origin of bee pollen samples.

Geographical Location	Sample Code	Visual Color	Family	Relative Frequency (%) of Pollen Types	Classification
Larache, MR	BP1	Light purple	Apiaceae	*Coriandrum* and *Daucus* sp. (100%)	Monofloral
Khenichat, MR	BP2	Yellow	Brassicaceae	*Brassica* sp. (60%), *Sinapis* sp. (30%) and *Tamarix* sp. (˂10%)	Polyfloral
Had Kourt, MR	BP3	Orange	Asteraceae	*Carduus*/*Galactites* sp. (35%), *Taraxacum* sp. (17%)*, Scorzonera/Lactuca* sp. (8%)*, Bellis* sp. (8%)*, Olea europea* (8%) and *Echium* sp. (6%), *Eucalyptus* sp. (3%)	Polyfloral
Kenitra, MR	BP4	Dark yellow	Oleaceae	*Olea europaea* (˃85%), *Tamarix* sp. (˂5%)	Monofloral
Fez, MR	BP5	Dark yellow	Brassicaceae	*Raphanus* sp. (˃80%) and *Sinapis* sp. (˂10%)	Monofloral
Sefrou, MR	BP6	Orange	Cistaceae	*Helianthemum* sp. (˃70%) and *Anthemis* sp. (˂10%), *Lhytrum* sp. (˂5%)	Polyfloral
Arfoud, MR	BP7	Light yellow	Fabaceae	*Ononis spinosa/Astralagus* sp. (˃90%), *Lhytrum* sp. and *Quercus* sp. (˂10%)	Monofloral
Taza, MR	BP8	Yellow	Fabaceae	*Ononis spinosa/Astralagus* sp. (˃90%), *Lhytrum* sp. and *Quercus* sp. (˂10%)	Monofloral

MR: Morocco, BP: bee pollen.

**Table 2 molecules-28-00835-t002:** Phenolic and phenylamide profile of Moroccan bee pollen samples. The values are expressed as mg of each compound/g of bee pollen.

Peak	Proposed Compound	BP1	BP2	BP3	BP4	BP5	BP6	BP7	BP8
1	Caffeic acid hexoside	0.36 ± 0.40	nd	nd	nd	0.14 ± 0.00	nd	nd	nd
2	Caffeic acid	nd	nd	nd	nd	nd	nd	nd	0.15 ± 0.00
3	*p*-coumaric acid hexoside	0.22 ± 0.02	0.16 ± 0.00	nd	nd	nd	nd	nd	nd
4	Myricetin-3-*O*-rutinoside	nd	nd	0.17 ± 0.00	nd	nd	0.80 ± 0.01	nd	nd
5	Quercetin-*O*-diglucoside	0.98 ± 0.02	1.77 ± 0.01	0.20 ± 0.00	3.30 ± 0.12	1.15 ± 0.10	0.53 ± 0.21	0.74 ± 0.00	0.43 ± 0.00
6	Myricetin-*O*-acetyl deoxyhexosyl-hexoside	nd	nd	nd	nd	nd	0.18 ± 0.01	nd	nd
7	Methylherbacetin-*O*-dihexoside	nd	0.17 ± 0.00	0.18 ± 0.01	nd	nd	nd	nd	nd
8	Myricetin-*O*-hexoside	nd	nd	nd	nd	nd	0.20 ± 0.01	nd	nd
9	Quercetin-*O*-pentosyl-hexoside	nd	nd	nd	nd	nd	nd	nd	0.17 ± 0.00
10	Quercetin-3-*O*-rutinoside	nd	nd	0.21 ± 0.00	0.38 ± 0.01	0.33 ± 0.06	0.48 ± 0.02	nd	nd
11	Kaempferol-*O*-diglucoside	nd	0.74 ± 0.00	nd	nd	0.83 ± 0.01	nd	0.30 ± 0.04	0.22 ± 0.00
12	Isorhamnetin-*O*-deoxyhexosyl-*O*-hexoside	nd	nd	nd	0.24 ± 0.01	nd	nd	nd	nd
13	Myricetin-*O*-malonyl hexoside	nd	nd	nd	nd	nd	0.24 ± 0.00	nd	nd
14	Methylherbacetin-3-*O*-rutinoside	nd	nd	0.18 ± 0.00	nd	nd	nd	nd	0.17 ± 0.00
15	Kaempferol-*O*-deoxyhexosyl-*O*-hexoside	nd	nd	nd	nd	0.22 ± 0.01	0.32 ± 0.01	nd	nd
16	Isorhamnetin-*O*-pentosyl-hexoside	nd	nd	0.18 ± 0.01	nd	nd	nd	0.17 ± 0.00	0.17 ± 0.00
17	Isorhamnetin-*O*-pentosyl-hexoside (isomer)	nd	nd	nd	nd	nd	nd	nd	0.20 ± 0.00
18	*p*-coumaric acid	0.20 ± 0.00	nd	nd	nd	nd	nd	nd	nd
19	Quercetin-*O*-malonyl deoxyhexosyl-hexoside	nd	nd	nd	nd	nd	0.23 ± 0.03	nd	nd
20	Kaempferol-3-*O*-rutinoside	nd	nd	0.17 ± 0.01	nd	0.30 ± 0.05	0.29 ± 0.00	0.23 ± 0.01	0.18 ± 0.00
21	Isorhamnetin-3-*O*-hexosyl-deoxyhexoside	nd	nd	nd	nd	nd	nd	nd	0.81 ± 0.20
22	Quercetin-3-*O*-glucoside	nd	nd	nd	0.35 ± 0.00	nd	0.19 ± 0.00	nd	nd
23	Isorhamnetin-*O*-malonyl rutinoside	nd	nd	0.21 ± 0.00	nd	nd	nd	nd	nd
24	Isorhamnetin-*O*-malonyl pentosyl-hexoside	nd	nd	0.28 ± 0.00	nd	nd	nd	nd	nd
25	Quercetin-*O*-malonyl hexoside	nd	nd	0.22 ± 0.00	nd	nd	0.84 ± 0.00	nd	nd
26	3’,4’,5’,3,5,6,7-heptahydroxy-flavonol-*O*-malonyl hexoside	nd	nd	nd	nd	nd	0.18 ± 0.00	nd	nd
27	Quercetin-*O*-malonyl hexoside (isomer)	nd	nd	nd	nd	nd	0.17 ± 0.00	nd	nd
28	Quercetin-3-*O*-rhamnoside	0.20 ± 0.01	nd	nd	nd	nd	1.19 ± 0.00	nd	nd
29	Isorhamnetin-3-*O*-glucoside	nd	nd	0.20 ± 0.00	nd	nd	0.20 ± 0.00	nd	0.16 ± 0.00
30	Kaempferol-*O*-malonyl rutinoside	nd	nd	0.19 ± 0.01	nd	nd	0.25 ± 0.01	nd	nd
31	Isorhamnetin-*O*-malonyl hexoside	nd	nd	0.26 ± 0.00	nd	nd	0.37 ± 0.01	nd	nd
32	Kaempferol-3-*O*-rhamnoside	1.60 ± 0.01	nd	0.21 ± 0.01	0.38 ± 0.03	nd	nd	nd	0.17 ± 0.00
33	Quercetin-3-*O*-rhamnoside	0.20 ± 0.01	nd	nd	nd	nd	1.19 ± 0.00	nd	nd
34	*N¹-p*-coumaroyl-*N⁵*, *N¹⁰*-dicaffeoylspermidine	nd	nd	nd	nd	0.24 ± 0.06	0.16 ± 0.00	0.36 ± 0.01	nd
35	*N¹*, *N⁵*, *N¹⁰*-*tri*-*p*-coumaroylspermidine	nd	nd	0.15 ± 0.00	nd	0.18 ± 0.00	nd	nd	nd
36	Kaempferol	nd	nd	nd	nd	nd	0.30 ± 0.00	2.80 ± 0.03	0.42 ± 0.00
37	Isorhamnetin	nd	nd	nd	nd	nd	0.19 ± 0.00	1.57 ± 0.08	0.15 ± 0.00
38	*N¹*, *N⁵*-*di*-*p*-coumaroyl-*N¹⁰*-caffeoylspermidine	nd	nd	nd	nd	nd	nd	0.51 ± 0.02	nd
39	*N¹*, *N⁵*, *N¹⁰*-*tri*-*p*-coumaroylspermidine (isomer)	nd	nd	nd	nd	0.25 ± 0.00	0.23 ± 0.00	2.48 ± 0.01	0.24 ± 0.00
40	*N¹*, *N⁵*, *N¹⁰*-*tri*-*p*-coumaroylspermidine (isomer)	nd	nd	nd	nd	0.19 ± 0.02	0.19 ± 0.00	1.61 ± 0.01	0.17 ± 0.00
41	*N¹*, *N⁵*, *N¹⁰*-*tri*-*p*-coumaroylspermidine (isomer)	nd	nd	nd	nd	nd	nd	0.61 ± 0.02	0.15 ± 0.00
42	*N¹*, *N⁵*, *N¹⁰*-*tri*-*p*-coumaroylspermidine (isomer)	nd	nd	0.19 ± 0.06	nd	0.32 ± 0.00	0.34 ± 0.00	10.52 ± 0.11	0.22 ± 0.00
43	Tetracoumaroyl spermine	nd	nd	0.20 ± 0.04	nd	nd	0.15 ± 0.01	nd	nd
44	Tetracoumaroyl spermine (isomer)	nd	nd	0.23 ± 0.06	nd	nd	0.22 ± 0.04	nd	nd
45	Tetracoumaroyl spermine (isomer)	nd	nd	0.14 ± 0.00	nd	nd	nd	nd	nd
46	Tetracoumaroyl spermine (isomer)	nd	nd	0.17 ± 0.01	nd	nd	nd	nd	nd
47	Tetracoumaroyl spermine (isomer)	nd	nd	0.16 ± 0.02	nd	nd	nd	nd	nd
48	Tetracoumaroyl spermine (isomer)	nd	nd	0.16 ± 0.02	nd	nd	0.16 ± 0.01	nd	nd
	Total phenolic acids (mg/g)	0.78	0.16	–	–	0.14	–	–	0.15
	Total flavonoids (mg/g)	2.98	2.68	2.86	4.65	2.83	8.82	5.81	3.25
	Total phenylamide derivatives (mg/g)	–	–	1.40	–	1.18	1.45	16.09	0.78

**Table 3 molecules-28-00835-t003:** Identification and quantification of volatile compounds in Moroccan bee pollen samples. The values are expressed as the relative percentage (R%).

Peak	Compound	BP1	BP2	BP3	BP4	BP5	BP6	BP7	BP8
1	2-propenylidene-cyclobutene	nd	nd	nd	nd	4.3 ± 0.7	nd	nd	4.7 ± 1.5
2	Hexanal	5.3 ± 0.8	14.92 ± 2.5	nd	nd	nd	12.3 ± 1.4	21.0 ± 2.3	59.8 ± 5.9
3	2-hexenal	2.2 ± 0.7	nd	nd	nd	nd	nd	5.5 ± 0.3	8.5 ± 2.4
4	Heptanal	nd	nd	nd	nd	nd	nd	nd	6.2 ± 0.3
5	2,5-dimethyl-pyrazine	3.9 ± 1.8	nd	nd	nd	nd	nd	nd	nd
6	1,2-cyclopentanedione	nd	nd	nd	0.9 ± 0.0	nd	nd	nd	nd
7	2,4-heptadienal	8.5 ± 2.9	33.6 ± 12.1	nd	nd	nd	nd	11.5 ± 1.2	nd
8	Ethyl hexanoate	nd	nd	nd	nd	nd	5.0 ± 0.8	nd	nd
9	Octanal	16.6 ± 1.3	nd	nd	nd	nd	nd	nd	nd
10	2,4-heptadienal (isomer)	9.2 ± 0.3	nd	nd	nd	nd	nd	nd	nd
11	Hexanoic acid	nd	nd	20.3 ± 3.4	nd	20.5 ± 2.3	nd	nd	nd
12	Eucalyptol	nd	10.6 ± 1.6	nd	nd	nd	6.4 ± 0.5	nd	nd
13	3,5-octadien-2-one	22.9 ± 2.1	27.5 ± 1.8	nd	nd	2.6 ± 1.2	12.4 ± 1.3	23.9 ± 3.4	nd
14	2,6,6-trimethylbicyclo [3.1.1]hept-3-ylamine	5.9 ± 1.0	nd	nd	nd	nd	nd	nd	nd
15	3,5-octadien-2-one (isomer)	12.7 ± 3.6	nd	nd	nd	nd	nd	25.6 ± 1.0	nd
16	Nonanal	1.9 ± 0.6	nd	nd	1.3 ± 0.1	nd	nd	1.7 ± 0.2	5.1 ± 1.1
17	Cis-β-terpineol	nd	nd	nd	nd	8.3 ± 0.8	nd	nd	nd
18	Methyl octanoate	nd	nd	nd	nd	nd	nd	1.4 ± 0.1	nd
19	Lilac aldehyde D	1.4 ± 0.1	nd	nd	nd	nd	nd	nd	nd
20	2,6-nonadienal	nd	nd	nd	nd	nd	nd	nd	3.0 ± 0.3
21	Isopinocarveol	nd	nd	nd	nd	nd	nd	1.3 ± 0.5	nd
22	Octanoic acid	4.3 ± 0.1	7.4 ± 0.6	2.9 ± 0.7	5.2 ± 1.4	7.5 ± 1.4	7.7 ± 0.8	1.9 ± 0.4	nd
23	Ethyl octanoate	nd	5.9 ± 2.0	14.8 ± 2.7	3.3 ± 0.2	nd	nd	1.2 ± 0.4	nd
24	Lilac alcohol D	3.8 ± 0.7	nd	nd	nd	nd	nd	nd	nd
25	β-cyclocitral	nd	nd	nd	nd	2.7 ± 0.6	nd	nd	nd
26	Methyl 7-hexanoate	nd	nd	nd	nd	nd	1.4 ± 0.4	nd	nd
27	Methyl nonanoate	nd	nd	nd	nd	nd	nd	1.5 ± 0.5	nd
28	Anisaldehyde	nd	nd	nd	5.7 ± 1.4	nd	nd	nd	nd
29	Geranyl vinyl ether	nd	nd	0.5 ± 0.0	nd	nd	nd	nd	nd
30	3-cyclohex-1-enyl-prop-2-enal	1.3 ± 0.2	nd	nd	nd	nd	nd	nd	nd
31	2-methyl-1-nonene-3-ine	nd	nd	nd	nd	nd	nd	nd	3.0 ± 0.2
32	Ethyl nonanoate	nd	nd	3.1 ± 1.2	3.3 ± 0.2	7.41 ± 2.00	nd	1.2 ± 0.8	nd
33	Nonanoic acid	nd	nd	nd	nd	8.22 ± 2.50	nd	1.6 ± 0.2	nd
34	Methyl 8-methyl-nonanoate	nd	nd	nd	nd	nd	nd	0.7 ± 0.1	nd
35	3-methyl-2-pent-2-enyl-cyclopent-2-enone	nd	nd	nd	2.4 ± 0.7	nd	nd	nd	nd
36	Ethyl decanoate	nd	nd	nd	nd	nd	10.9 ± 2.5	nd	nd
37	Methyl octanoate	nd	nd	11.5 ± 3.0	13.1 ± 2.3	nd	11.8 ± 2.1	nd	nd
38	Caryophyllene	nd	nd	nd	3.8 ± 0.3	nd	nd	nd	nd
39	Decanoic acid	nd	nd	nd	nd	nd	22.8 ± 1.3	nd	nd
40	6,10-dimethyl-5,9-undecadien-2-one	nd	nd	nd	8.4 ± 1.4	17.6 ± 2.8	nd	nd	nd
41	4,6-dimethyl-(Z)-5,9-undecadien-2-one	nd	nd	5.7 ± 0.2	nd	nd	nd	nd	nd
42	β-ionone	nd	nd	nd	nd	1.6 ± 0.3	nd	nd	nd
43	β-ionone epoxide	nd	nd	nd	nd	11.7 ± 0.1	nd	nd	nd
44	10-methyl-methyl undecanoate	nd	nd	1.1 ± 0.5	nd	nd	nd	nd	nd
45	Ethyl decanoate	nd	nd	16.7 ± 4.3	1.4 ± 0.4	nd	nd	nd	nd
46	Ethyl dodecanoate	nd	nd	nd	nd	7.4 ± 1.8	nd	nd	nd
47	5-(1-piperidyl)-furan-2-carboxaldehyde	nd	nd	2.7 ± 1.4	nd	nd	nd	nd	nd

## Data Availability

Data is contained within the article and Appendix A.

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
