# Peer review of "Evaluation of Antioxidant and Anticancer Activity of Mono- and Polyfloral Moroccan Bee Pollen by Characterizing Phenolic and Volatile Compounds"

_molecules, 2023, doi:10.3390/molecules28020835_

Round 1

Reviewer 1 Report

The authors reported the potential biological properties of bee pollen. The manuscript is well written and data is well presented. My only recommendation is to improve the discussion on the correlations between the antioxidant antivity and the content of total phenolics. The authors can use the following article as guide for the discussion: https://doi.org/10.1111/j.1750-3841.2009.01352.x

Author Response

Reviewer #1:

The authors reported the potential biological properties of bee pollen. The manuscript is well written and data is well presented.

  1. My only recommendation is to improve the discussion on the correlations between the antioxidant activity and the content of total phenolics. The authors can use the following article as guide for the discussion: https://doi.org/10.1111/j.1750-3841.2009.01352.x

Our response: We thank the reviewer suggestion, and following the recommendations we improve the discussion to correlate activity and phenolic composition.

Reviewer 2 Report

The work by Volkan Aylanc and colleagues shows the antioxidant and anti-tumor potential of mono- and polyfloral bee pollen samples. Some samples revealed a moderate effect on cervical carcinoma and breast adenocarcinoma cell lines. Overall, the results highlighted the potentiality of bee pollen to serve in food supplements and health-promoting formulations in the future.

I have found some minor details in the ms that need to be revised, hopefully improving the present work's quality.

1)           Keywords: Please do not include words mentioned in the title of the ms, and Sort key-words alphabetically. 2)           In the description of the figures, abbreviations should not be used.

3)               Why did you choose some Moroccan locations and not others concerning the botanical origin of pollen?

Author Response

I have found some minor details in the ms that need to be revised, hopefully improving the present work's quality.

  1. Keywords: Please do not include words mentioned in the title of the ms, and Sort key-words alphabetically.

Our response: Changes were made following the reviewer suggestion.

  1. In the description of the figures, abbreviations should not be used.

Our response: Changes were made following the reviewer suggestion.

Reviewer 3 Report

The study is of excellent quality and so is the manuscript. My only significant comment is that Tables 2 and 3 are too dense and difficult to read. I strongly advise that the authors split both tables into an SI section and a manuscript section. Specifically, table 2, columns 1, 2-6 should be an SI table and columns 1,6 -14 should appear in the main manuscript. This would provide more room so that the BP1-8 data are all on single lines and thus easier to read.  The same should be done for Table 3, i.e., place the BP1-8 data on single lines so it is easier to read.

Also, neither table 2 or 3, or their legends, contains any indication of units for the BP1-8 data. Please clearly indicate how to interpret these values in the table column headers (units) or the legend (explanation).

Author Response

  1. The study is of excellent quality and so is the manuscript. My only significant comment is that Tables 2 and 3 are too dense and difficult to read. I strongly advise that the authors split both tables into an SI section and a manuscript section. Specifically, table 2, columns 1, 2-6 should be an SI table and columns 1,6 -14 should appear in the main manuscript. This would provide more room so that the BP1-8 data are all on single lines and thus easier to read. The same should be done for Table 3, i.e., place the BP1-8 data on single lines so it is easier to read.

Our response: Changes on the tables were made accordingly to the reviewer suggestion. Additional information was sent to supplementary material.

  1. Also, neither table 2 or 3, or their legends, contains any indication of units for the BP1-8 data. Please clearly indicate how to interpret these values in the table column headers (units) or the legend (explanation).

Our response: Changes were made following the reviewer suggestion, adding the units in the table header.

Reviewer 4 Report

The paper reports a systematic evaluation of the antioxidant and anticancer activity of mono- and poly-floral Moroccan bee pollen with respect to their phenolic and flavonoid content and their volatile compounds. The paper falls within the scope of the journal and might be of interest for its readers. The presentations is clear, although Figure 3 is not reproduced properly.  

Author Response

The paper reports a systematic evaluation of the antioxidant and anticancer activity of mono- and poly-floral Moroccan bee pollen with respect to their phenolic and flavonoid content and their volatile compounds. The paper falls within the scope of the journal and might be of interest for its readers.

  1. The presentations is clear, although Figure 3 is not reproduced properly.

Our response: We improved the quality of figure 3, as recommended by the reviewer. Nevertheless, we will be able to provide the figure in a different format, if required, during the publication process.

Reviewer 5 Report

This paper evaluates the antioxidant and anticancer activity of mono- and polyfloral Moroccan bee pollen by characterizing phenolic and volatile compounds. However, a major revision should be done before future consideration.

-Analysis methods and studied characteristics/tests should be presented in the abstract. Which health and biological characteristics?? Please clarify.

- The abstract should be restructured carefully.

- The novelty of the study should be highlighted in the introduction compared to other previous studies.

-It is advised to authors to not to use the abbreviations in the abstract section. If necessary, please define them at their 1st use.

- Which extracted volatile compounds and bioactive compounds had the greatest anticancer and antioxidant effects? They should be highlighted in the abstract and conclusion

- Figure 5, how many replicates did you perform? This information with error bars should be provided.

- The X and Y axis of the Figures are hardly visible. Please choose a bigger font to make the Figures overall look more visible.

Author Response

This paper evaluates the antioxidant and anticancer activity of mono- and polyfloral Moroccan bee pollen by characterizing phenolic and volatile compounds. However, a major revision should be done before future consideration.

  1. Analysis methods and studied characteristics/tests should be presented in the abstract. Which health and biological characteristics?? Please clarify.

Our response: Changes were made following the reviewer suggestion.

  1. The abstract should be restructured carefully.

Our response: The abstract was reorganized following the reviewer suggestions.

  1. The novelty of the study should be highlighted in the introduction compared to other previous studies.

Our response: Changes were made following the reviewer suggestion.

  1. It is advised to authors to not to use the abbreviations in the abstract section. If necessary, please define them at their 1st use.

Our response: The abbreviations were removed from the abstract with exception of the chemicals used on the antioxidant activity, since they are commonly known by their abbreviation.

  1. Which extracted volatile compounds and bioactive compounds had the greatest anticancer and antioxidant effects? They should be highlighted in the abstract and conclusion.

Our response: Changes were made in the abstract and conclusion to highlight the compounds that may be contribute most for the bioactivity, as suggested by the reviewer.

  1. Figure 5, how many replicates did you perform? This information with error bars should be provided.

Our response: The number of replicates is given in section ''3.9. Statistical Analysis'' and were done triplicate. Samples with GI > 1000 were calculated as 1000 and hence the standard deviation values were zero, but the error bars for sample BP4 (HeLa) and BP1 (MCF-7) is given in the figure.

  1. The X and Y axis of the Figures are hardly visible. Please choose a bigger font to make the Figures overall look more visible.

Our response: Changes were made, for all figures, following the reviewer suggestion.

Round 2

Reviewer 5 Report

Accept in present form

Author Response

We checked the manuscript for minor errors and made some additional improvements. There are no additional specific comments from reviewer 5.